# Assessment of Postural Control in Children with Movement Disorders by Means of a New Technological Tool: A Pilot Study

**DOI:** 10.3390/bioengineering11020176

**Published:** 2024-02-11

**Authors:** Valentina Menici, Roberta Scalise, Alessio Fasano, Egidio Falotico, Nevio Dubbini, Giuseppe Prencipe, Giuseppina Sgandurra, Silvia Filogna, Roberta Battini

**Affiliations:** 1Department of Developmental Neuroscience, IRCCS Fondazione Stella Maris, 56128 Pisa, Italy; valentina.menici@fsm.unipi.it (V.M.); roberta.scalise@fsm.unipi.it (R.S.); silvia.filogna@fsm.unipi.it (S.F.); roberta.battini@fsm.unipi.it (R.B.); 2Ph.D. Programme in Clinical and Translational Sciences, University of Pisa, 56126 Pisa, Italy; 3The BioRobotics Institute, Scuola Superiore Sant’Anna, 56127 Pisa, Italyegidio.falotico@santannapisa.it (E.F.); 4Department of Excellence in Robotics and AI, Scuola Superiore Sant’Anna, 56127 Pisa, Italy; 5IRCCS Fondazione Don Carlo Gnocchi ONLUS, 50143 Florence, Italy; 6Miningful srls, 56121 Pisa, Italy; neviod@miningfulstudio.eu; 7Department of Computer Science, University of Pisa, 56127 Pisa, Italy; giuseppe.prencipe@unipi.it; 8Department of Clinical and Experimental Medicine, University of Pisa, 56126 Pisa, Italy

**Keywords:** diagnostic technological tools, balance, gait impairments, movement disorder, children

## Abstract

Considering the variability and heterogeneity of motor impairment in children with Movement Disorders (MDs), the assessment of postural control becomes essential. For its assessment, only a few tools objectively quantify and recognize the difference among children with MDs. In this study, we use the Virtual Reality Rehabilitation System (VRRS) for assessing the postural control in children with MD. Furthermore, 16 children (mean age 10.68 ± 3.62 years, range 4.29–18.22 years) were tested with VRRS by using a stabilometric balance platform. Postural parameters, related to the movements of the Centre of Pressure (COP), were collected and analyzed. Three different MD groups were identified according to the prevalent MD: dystonia, chorea and chorea–dystonia. Statistical analyses tested the differences among MD groups in the VRRS-derived COP variables. The mean distance, root mean square, excursion, velocity and frequency values of the dystonia group showed significant differences (*p* < 0.05) between the chorea group and the chorea–dystonia group. Technology provides quantitative data to support clinical assessment: in this case, the VRRS detected differences among the MD patterns, identifying specific group features. This tool could be useful also for monitoring the longitudinal trajectories and detecting post-treatment changes.

## 1. Introduction

### 1.1. Movement Disorders in Children: Definition and State of Art

Movement Disorders (MDs) in children include a highly heterogeneous group of conditions that lead to the impairment of motor ability, dysfunction of postures, or abnormal involuntary movements, starting acutely, subacutely, or chronically. MDs can be extremely disabling in the pediatric population and disruptive to psychomotor development [1,2]. Childhood MDs are often classified into two main categories: hyperkinetic/dyskinetic MDs (including dystonia, chorea, tremor, myoclonus, tics and stereotypies) [3] and hypokinetic movement disorders, encompassing parkinsonian phenotypes or akinetic/rigid disorders [4].

Hyperkinetic movements, as dystonia and chorea, are “any unwanted excess movement” [2], and their phenomenology should be described in terms of duration, velocity, amplitude, rhythmicity, jerkiness, repeatability, or stereotyped quality and the number of different identifiable movements or postures.

Referring to hyperkinetic/dyskinetic MDs, dystonia is characterized by sustained or intermittent muscle contractions, often repetitive, causing abnormal movements and postures or both [3,4]. In particular, dystonic movements typically involve twisting, are patterned and abrupt, and may be tremulous [2]. Instead, chorea is an MD characterized by a random-appearing sequence of one or more discrete involuntary movements or movement fragments that may flow from one part of body to another [3]. Chorea movements are usually irregular, random (due to variability in timing, duration rate, direction, or anatomic location), chaotic and rapid. This is why they are often described as “dance-like” [2]. Moreover, dystonia and chorea in children with MD can be expressed independently or combined; indeed, as confirmed by the literature, in Dyskinetic Cerebral Palsy (DCP), they are frequently co-occurring [5].

Children with MD show dysfunctional postural control, impeding their achievement of motor milestones related to independent stance and the performance of new motor skills. This impairment also affects their functional performance in daily life activities. Consequently, it is crucial to identify and assess the balance control of children with MD to establish a personalized and tailored treatment plan, whether it involves rehabilitation or pharmacological and/or surgical intervention. Additionally, the monitoring and tracking of movements are essential. Therefore, there is a growing need for an adequate assessment tool for postural control in children with MDs.

Postural control in children with MD is highly variable and diverse depending on the specific type of MD. Balance skills play a crucial role in the development of gross motor abilities and significantly impact the overall quality of life. Hence, there is a necessity to establish standardized outcome measures for the evaluation and quantification of postural control; however, this task can be challenging in children, particularly when various motor patterns are present.

Currently, instruments and rating scales are employed in clinical practice to assess MD in children; however, most of them has been designed for use with adults. In current clinical practice, instruments and rating scales are employed to assess the severity of MD, global motor functions, and the impact on daily life activities. However, most of these tools are designed for being used in adults (i.e., the Fahn–Marsden (BFM) Scale, the Unified Huntington’s Disease Rating Scale, the Barry–Albright Dystonia Scale) [6,7,8,9] and only a few were designed specifically for children (i.e., the Movement Disorder Childhood Rating Scale 0–3 and 4–18 Revised and the Dyskinesia Impairment Scale) [10,11]. Although these scales measure the severity of MD, specific global motor functions and how the MD impacts in the motor abilities and in daily life activity, none of them specifically evaluate postural and balance capabilities.

Children with MD often exhibit dysfunctional postural control, hindering the attainment of motor milestones and affecting daily life activities. Postural control in children with MD is highly variable and diverse depending on the specific type of MD. Balance skills play a crucial role in the development of gross motor abilities and significantly impact the overall quality of life. Hence, there is a necessity to establish standardized outcome measures for the evaluation and quantification of postural control; however, this task can be challenging in children, particularly when various motor patterns are present.

Despite available tests for measuring balance in children in clinical practice, such as developmental scales with balance subscales (e.g., balance subtest of the Bruininks–Oseretsky test, second edition, or balance subsection scores of Movement ABC-2 or Praxic and Motor Coordination Skills APCM-2) or exclusive balance measurement tools with either functional balance tests (e.g., the Timed-Up and Go test (TUG), Lower Quarter Y-Balance test (YBT-LQ), Berg Balance Scale (BBS), Pediatric Balance Scale (PBS) and Functional Reach Test (FRT)) [12,13,14,15,16], none are designed specifically for MDs, and the quantitative data obtained from existing scales are operator-dependent. However, these tools are not designed specifically for MDs. All these scales provide quantitative data obtained from the score assigned by the clinician, making them operator- and experience-dependent. While they can serve as outcome measures for detecting changes, instrumented quantitative technological methods can be more objective, sensitive, and reliable.

### 1.2. Postural Control Assessment and Treatment with Technology: State of the Art

A quantitative assessment of postural parameters derived from the trajectories of the Centre of Pressure (COP) can be obtained by a high-precision force plate which is considered the gold standard of balance performance during quiet and/or perturbed standing [17]. Specifically, the COP represents the point of application of the ground reaction force vector of the supporting surface, and it enables the analysis and quantification of postural stability and evaluation of different postural strategies [18]. By assessing the displacement of the COP, one can analyze the postural strategies used to maintain a standing position in both the anteroposterior (AP) and mediolateral (ML) directions, specifically focusing on movements involving ankles and hips [19,20,21].

There are many technologies such as virtual reality applications, telerehabilitation systems, web-based applications and robotic systems for evaluating balance; in addition wearable sensors, alone or combined with other devices, can be used, leading to flexibility in clinical and research measurements [17,18].

To obtain detailed postural analysis and evaluate strategies underlying balance mechanisms, the use of technologically instrumented tests can be employed specifically during the developmental age to make the evaluation less boring for children. Indeed, the “exergaming”, a combination of ‘exercise’ and ‘gaming’, is a term used to describe computer applications that require physical activity in response to game demands. Exergaming is a promising new alternative to traditional modes of therapeutic exercise which may be preferable and more effective both for healthy and neurologically impaired populations [22].

In this context, positive results were obtained by using technological devices in older adults [23], patients with stroke [24,25], Cerebral Palsy (CP) [26], Parkinson’s disease [27,28], and brain injury [29], allowing improvements in balance, gait and function. In the literature, many studies are present on the study of balance in Parkinson’s disease using different technologies for assessment and rehabilitation [30,31,32], but no studies report postural control or balance assessments in MD, in particular in children.

Commercial systems like XBOX Kinect™ and Nintendo Wii have been integrated into clinical practice to assess coordination and balance in school-aged typically developing children [30] and used as rehabilitation tools [31,32], but they are mainly produced for entertainment purposes. This technology is certainly attractive and engaging for children.

In this frame, recent studies have shown that a new technological and advanced tool, the Virtual Reality Rehabilitation System (VRRS), can be used for both the assessment [33,34] and rehabilitation/telerehabilitation of postural control [35,36] in children and adolescents.

The range of diseases targeted by the VRRS method is fairly wide. Although the system was originally conceived to give a response to neurological rehabilitation after stroke in the adult population [37,38], it is versatile enough to be used for neurological (tele-)rehabilitation of numerous diseases such as multiple sclerosis, traumatic brain injury, chronic pain and others [39,40,41]. In addition, it has been shown that an interactive environment yields optimal results regarding the motivation to reach the goals and to pursue a specific program and can be also useful for physically impaired children [42].

Based on the evidence that the VRRS system is able to discriminate against different diseases [33], the aim of this study is to use the quantitative parameters of the VRRS system to identify potential differences in balance abilities among three different types of MDs (i.e., dystonia, chorea, chorea–dystonia). Specifically, our aim is to determine whether the resulting differences identified will be in line with the clinical phenomenology and assessment of MD in children.

## 2. Materials and Methods

### 2.1. Participants

Children aged from 5 to 18 years admitted to the IRCCS Fondazione Stella Maris for clinical follow-up have been recruited according to the following criteria:-Presence of MD as dystonia, chorea, or chorea–dystonia, with different etiology;-Level I to III of the Gross Motor Function Classification System (GMFCS);-Exclusion criteria were:-GMFCS level > III;-Verbal Intelligence Quotient < 80, as assessed with the Wechsler Intelligence Scale for Children that prevented understanding the required tasks.

Socio-demographic information, such as age and gender, and clinical characteristics such as diagnostic etiology, were obtained from all study participants.

The current study has been approved by the Tuscany Pediatric Ethics Committee (221/2020).

Children were tested with the VRRS system according to a standardized assessment protocol.

### 2.2. VRRS System

The VRRS (Khymeia, Italy) is an innovative and modular system for rehabilitation and telerehabilitation based on the concept of augmented feedback. It is a fully functional architecture that allows for neurological, cognitive, postural, speech and orthopedic rehabilitation activities in VR environments using different peripherals, including a stabilometric balance. VRRS is conceived to treat the subject by generating augmented feedback toward his central nervous system through targeted rehabilitation programs which are performed in a virtual environment that helps the subjects develop knowledge of the results of the movements [43,44]. There are several packages for rehabilitation and others for assessments.

The stabilometric balance is a force plate (80 × 55 cm) able to detect the COP displacement from forces in the z-direction. Specifically, the four load cells integrated into the balance platform can detect the forces applied, and, from these, the system is able to compute the anterior–posterior (AP) and medial–lateral (ML) components of the COP. An internal dedicated software can estimate static postural parameters.

The data obtained for each exercise during the assessment with VRRS have been exported and gathered in reports.

### 2.3. VRRS Balance Assessment

According to the International Posture and Gait Research Society recommendation, a specific protocol was considered. It is important to have stable and reliable parameters within the recording time of 25–40 s under static conditions, since the steady process of posture control requires a few seconds of adjustment time [45,46].

Furthermore, it was crucial that the test did not induce fatigue, and the subject, before starting the registration, could choose the posture of the foot and adjust it without stepping off the force plate. Guidelines were also given for the target distance to be observed during the test that was placed at a height equivalent to the subject’s line of sight and at a distance between 1 and 3 m [20].

Considering these indications, an ad hoc protocol for the evaluation of the subjects enrolled in the present study were implemented. Children were asked to keep the standing position independently on the VRRS stabilometric balance platform for 60 s, with their eyes open, looking at a fixed dot on the screen (Figure 1). This evaluation was repeated for 6 times, providing breaks between repetitions.

### 2.4. VRRS Data Collection

Subjects underwent a standardized number of trials of postural control assessment. Each trial lasted 60 s, and the VRRS system’s output data were averaged over the time series of the single test.

The parameters collected and analyzed in the present work were chosen on the basis of a previous study with the VRRS system [33], and they are also the most used for the assessment of stance stability.

In particular, these measures regarded the COP and its AP and ML components and are defined as follows:-Mean distance of the COP (MD_COP), AP (MD_AP) and ML (MD_ML), i.e., average displacement of the COP, AP and ML components from the central point of the stabilogram, respectively [mm];-Root mean square distance of the COP (RMS_COP), AP (RMS_AP) and ML (RMS_ML) from the central point of the stabilogram [mm];-Total path length (excursion) of the COP (ESC_COP), i.e., the sum of the distances between consecutive points on the COP path and its AP (ESC_AP) and ML (ESC_ML) components [mm];-Average velocity of the COP (VEL_COP), AP (VEL_AP) and ML (VEL_ML), defined as the total excursion divided by an analysed temporal interval (60 s) [mm/s];-Sway area enclosed by the COP path per unit of time (SWAY) [mm^2^/s];-Mean rotational frequency of the COP (FREQ_COP), AP (FREQ_AP) and ML (FREQ_ML) [Hz].

### 2.5. Statistical Analysis

Separate statistical modelling was carried out for each outcome, i.e., the above-mentioned VRRS-derived COP variables. Statistical analyses aimed to identify statistically significant differences among MD groups in the VRRS-derived COP variables. For each outcome, a number of outliers ranging from 1 to 4 were identified and excluded from the analysis with the Mahalanobis distance. Afterward, linear mixed models were run, with the MD group, age and therapy, together with interactions as fixed factors and the patient as the random factor (random intercept). The statistical significance was set at 0.05. Analyses were carried out with R (version 4.3.1, 2023-06).

## 3. Results

### 3.1. Participants

A total of 16 children participated in the study, with a mean age of 10.68 ± 3.62 years (range 4.29–18.22 years). The sample was divided into three MD groups, dystonia, the chorea and chorea–dystonia group, and they were composed as follows: six children with dystonia (37%), five with chorea (31.5%) and five with chorea–dystonia (31.5%).

Moreover, six subjects (37.5%) were affected by dyskinetic cerebral palsy, and the remaining ten (62.5%) were affected, seven by genetic syndromes and three still unknown.

The number of children undergoing pharmacological therapy in total was eight, specifically, two for the dystonia group, four for the chorea group, and two for the chorea–dystonia group.

All participants completed the assessment maintaining the standing position for the specified length of time.

### 3.2. VRRS-Derived COP Variables

All data were collected without missing values. Table 1 shows coefficients of the linear mixed model (estimate) and the associated *p*-values for each outcome. The same information for interactions is reported in Table 2.

The MD_AP value is significantly different between the chorea and dystonia group (*p* = 0.04). Indeed, the chorea group has a lower mean MD_AP value than the dystonia group (Figure 2).

The RMS_AP value is significantly different between the chorea and dystonia group (*p* = 0.02). Indeed, the chorea group has a lower mean RMS_AP value than the dystonia group (Figure 3).

Subjects who are taking therapy show a significant difference in RMS_AP values compared to the others (*p* = 0.04). The interaction between the chorea group and therapy is significant (*p* = 0.03): subjects in the chorea group undergoing therapy have a higher RMS_AP value than the dystonia group, on average. The same goes for the interaction between the chorea–dystonia group and therapy compared to the dystonia group (*p* = 0.02).

The interaction between the chorea group and age is significant for RMS_AP values (*p* = 0.03) because the difference in RMS_AP value between the chorea group and the dystonia group widens as age increases.

The interaction between the chorea–dystonia group and age is significant for ESC_COP values (*p* = 0.05), since in the chorea–dystonia group, as age increases, the ESC_COP value decreases more compared to dystonia group.

The VEL_AP value is significantly different between the chorea and dystonia group (*p* = 0.04). Indeed, the chorea group has a lower mean VEL-AP value than the dystonia group (Figure 4). The VEL_AP value is significantly different between the chorea–dystonia group and the dystonia group (*p* < 0.01). Indeed, the chorea–dystonia group has a higher mean VEL_AP value than the dystonia group (Figure 4).

The interaction between the chorea group and age is significant for VEL_AP values (*p* = 0.04), since in the chorea group, as age increases, the VEL_AP value also increases, compared to the dystonia group.

The interaction between the chorea–dystonia group and age is significant for the VEL_AP value (*p* < 0.01), since in the chorea–dystonia group, as age increases, the VEL_AP value decreases more compared to the dystonia group.

The interaction between the chorea–dystonia group and therapy is significant (*p* = 0.04): subjects in the chorea–dystonia group undergoing therapy have a higher VEL_AP value than the dystonia group, on average.

The FREQ_COP value is significantly different between the chorea and dystonia groups (*p* = 0.02). Indeed, the chorea group has a higher mean FREQ_COP value than the dystonia group (Figure 5).

The interaction between the chorea group and age is significant for the FREQ_COP value (*p* = 0.03), since in the chorea group, as age increases, the FREQ_COP decreases more compared to the dystonia group.

Both the interaction between the chorea group and therapy and the interaction between the chorea–dystonia group and therapy are significant (both *p*= 0.02): subjects in the chorea and chorea–dystonia groups undergoing therapy have a lower FREQ_COP value than the dystonia group, on average.

The FREQ_ML value is significantly different between the chorea and dystonia groups (*p* < 0.01). Indeed, the chorea group has a higher mean FREQ_ML value than the dystonia group.

The FREQ_ML value is significantly different between the chorea–dystonia and dystonia groups (*p* = 0.02). Indeed, the chorea–dystonia group has a higher mean FREQ_ML value than the dystonia group (Figure 6).

The interaction between the chorea group and age is significant for the FREQ_ML value, since in the chorea group, as age increases, the FREQ_ML decreases more compared to the dystonia group.

The interaction between the chorea group and therapy and the interaction between the chorea–dystonia group and therapy are significant (both *p* < 0.01): subjects in the chorea and chorea–dystonia groups undergoing therapy have a lower FREQ_ML value than the dystonia group, on average.

## 4. Discussion

This study aimed to demonstrate the applicability of using technology to describe and recognize the different patterns of movement in children with MDs, highlighting the most different features of the movement.

The quantitative analysis of different movement components allowed us to distinguish between various patterns, offering potentially valuable information for diagnosis, therapy, and treatment.

The use of technology provides several advantages for studying MDs. Firstly, it allows one to acquire a large amount of data in an objective and repeatable manner, minimizing the risk of human errors. Additionally, the use of advanced analysis techniques enables the identification of patterns that would not be possible with traditional methods.

The results obtained are in line with those of other studies that have used motion analysis techniques to study MDs. In particular, it has been shown that hypokinetic movement disorders, such as Parkinson’s disease, are characterized by a slowing of movement, while hyperkinetic MDs are characterized by an increase in the frequency and amplitude of movements [47].

Several studies in the literature have investigated balance abilities in typically developing children [48], but few have focused on children with specific motor difficulties such as developmental coordination disorder [49], CP [50,51,52], or benign cerebellar tumor [53].

More frequently, in fact, posturography is used to assess postural control, as in the Pierre et al. study that compared children with CP to typically developing children [52]. Balance abilities in stable sitting, unstable sitting, and quiet standing, under “eyes open” and “eyes closed” conditions were assessed. The results obtained show that children with CP have a specific impairment in the postural control of axial segments and this influences standing and walking, so it needs to be taken into account in rehabilitation programs for children with CP [52].

In this study, we have, for the first time, developed a standardized postural control assessment that is consistent across all subjects, utilizing a technological system. We also took into consideration the prevalent type of MD in each subject, thereby distinguishing between the three categories (dystonia, chorea, and chorea–dystonia).

We identified clinically relevant features specific to the dystonic component when compared to chorea or chorea–dystonia. Additionally, we analyzed potential interactions of these groups with the subject’s age or the use of pharmacological treatments.

In accordance with the literature, we considered the specific characteristics of the movement. Indeed, Singer and colleagues recommend a quantitative measurement of kinematics parameters such as position, velocity, excursion, frequencies and acceleration, facilitating an accurate diagnosis and detailed assessment of MDs [2].

While clinical descriptions have previously outlined differences in movement components, such as speed, amplitude of movement, and frequency, within patterns of MD, in this study we exploited the use of technological parameters to quantitatively analyze and collectively assess these various components of MD. Through this approach, we confirmed the existence of specific quantitative differences among these features.

Specifically, differences in the VRRS parameters between the dystonia and chorea groups highlight statistically significant variations. In the chorea group, the values for AP movement in velocity (VEL_AP) and mean distance (MD_AP) were statistically lower compared to the dystonia group, indicating reduced mean values. This finding is consistent with the literature, which describes dystonia movements as being abnormal, often repetitive, and characterized by patterned, twisting, and sometimes rhythmic motions. Therefore, dystonic movements are less frequent but faster and wider than chorea movements according to these results [1,2,50].

In addition, we observed from the obtained results that the older children with chorea MD showed a greater difference in AP velocity values compared to the dystonia group.

As for the frequency parameter of the COP (FREQ_COP) and the ML values (FREQ_ML), we observed that in the dystonia group, the values were lower compared with chorea, and this is in line with the clinical description of the different movement patterns because dystonia is distinguished from chorea based on the more predictable and stereotyped movements or postures compared with the apparently random, unpredictable, frequent and continuously ongoing nature of the movements of chorea [2].

An interesting result obtained from the interaction of the frequency values (both FREQ_CoP and FREQ_ML) of the chorea group with age showed that as age increases, the frequency values decrease compared to the dystonia group. This can be attributed to the fact that, as children grow, they adopt compensatory mechanisms to better control involuntary movements. They learn to anticipate and manage these involuntary movements, develop strategies to avoid, reduce or mitigate their impact, and adapt their movement patterns to accommodate these involuntary actions [54] [55].

Instead, for chorea–dystonia, characterized by both MDs, the pattern is a medium-to-higher velocity and the phenomenon of *geste antagoniste* (sensory trick) is very common. Typical for the dystonia component of movement, the maintenance of asymmetric “en bloc” postures on the transverse plane is used to fix the rapid component of the movement [2].

In addition, regarding the interaction between the chorea–dystonia group and age, it was observed that as age increases, the velocity (VEL_AP) and excursion (ESC_CoP) values also decrease compared to the dystonia group. This phenomenon can be elucidated by their capacity to formulate more robust *geste antagoniste*. Consequently, this mixed form exhibits a movement pattern closely resembling that of the dystonic group, as they employ fixation as a compensatory mechanism, as elucidated earlier.

Furthermore, the FREQ_COP values of the chorea–dystonia group were similar to those of the dystonia group when children were not taking therapy. However, when children were taking therapy, the FREQ_COP values of the chorea–dystonia group were significantly lower than the others. We hypothesize that this finding can be attributed to the fact that pharmacological therapy can help to reduce the involuntary movements caused by chorea. This reduction in involuntary movements can lead to a change in the movement pattern, which can be reflected in the FREQ_COP values [54].

Regarding the RMS_AP values, which represent a simple measure of deviation (variability) in the COP values recorded in the AP direction relative to its mean value during the test, the results confirmed that the chorea group is significantly different from the dystonia group. This significance remained between the two groups even when the subjects in the chorea group received therapy or if the age increased. Significant values for the value of RMS_AP were also obtained in the comparison between the dystonia group and the chorea–dystonia group but only by including the pharmacological treatment interaction in the analysis; this determines that the values between the two groups differ only if patients take therapy. According to the literature, Prieto et al. (1996) reported that the RMS measure of postural sway is a reliable measure of the magnitude of oscillation. The RMS measure is also believed to be correlated with the effectiveness/stability of the postural control system. Therefore, this confirms that the VRRS system provides reliable values [56].

Overall, these findings highlight the potential relation between the clinical phenomenology and the VRRS parameters related to different types of MD.

The strength of this study is that in using the technological assessment, we can obtain non-operator-dependent quantitative data through innovative proposals.

The obtained results serve multiple purposes: (i) enhancing the characterization of MDs because quantitative analysis of various movement components aids in distinguishing between different types of MDs, providing valuable diagnostic insights; (ii) fostering the development of innovative technology-based treatments in order to understand the underlying characteristics of MDs and facilitate the creation of more effective and engaging therapeutic approaches; and (iii) following the progression of MDs because quantitative movement analysis enables the longitudinal monitoring of these disorders, offering valuable information for assessing intervention effectiveness.

The postural control evaluation in the VRRS adheres to a standardized protocol with a defined test duration, consistent environmental conditions, and is based on exergaming. While low-cost, mass-produced technology systems like Wii and Xbox exist, they have technical limitations that may impact their validity due to the implemented filtering and data sampling techniques designed to minimize errors [57].

Despite the use of this innovative instrument, the study has limitations, including: (i) a restricted number of collaborative patients meeting the inclusion criteria; (ii) the absence of a detailed analysis of posturographic parameters for MD, especially for other diseases (e.g., post-stroke, multiple sclerosis, Parkinson’s, etc.); (iii) the absence of a test–retest component, such as reassessing the same child after a few days, which would confirm the validity and test the reliability of the assessment; (iv) the reliance on system-provided data, although raw data analysis (average values) generally allows for a more comprehensive assessment; and (v) a limited dataset to apply classification algorithms.

For future studies, it would be beneficial to compare data from clinical assessment scales with technological ones or explore alternative quantitative technological tools, especially for use in children with MDs. Additionally, the inclusion of data from age-matched typically developing children, could add insights for detecting different features between typical development and different MDs or other neurodevelopmental disorders.

Moreover, we plan to explore the application of machine learning algorithms to further refine the possibility to differentiate between the different types of MD in the pediatric population. Currently, the available dataset precludes the effective implementation of such algorithms; we will integrate a machine learning-based analysis as soon as we have access to a more extensive and representative dataset to significantly contribute to a more precise understanding and classification of MDs in children.

The achieved results mark a significant advancement in understanding and assessing postural control in children with both genetic and acquired MDs using non-operator-dependent data. The integration of technology offers new possibilities for enhancing our comprehension of these disorders and developing innovative methods for diagnosis, assessment and monitoring.

## 5. Conclusions

This study has suggested that the use of technology can be a valuable tool for improving the understanding of MDs in children.

This research can also assist clinicians in treating patients. The quantitative data obtained from the VRRS system enables clinicians to assess children with MDs before or after pharmacological intervention, monitoring their progress over time. Additionally, it allows the evaluation of balance skills before and after specific rehabilitation treatments.

Consequently, the VRRS can be a feasible instrument for the assessment of postural control in children with different types of MDs, showing promising for further development.

Our findings support the use of the VRRS system in the pediatric population due to its versatile technology, offering the capability to adapt activities in real-time and being well perceived by participants. It could represent a valuable, innovative tool that complements clinical rating scales, making it particularly useful in routine clinical settings.

## Figures and Tables

**Figure 1 bioengineering-11-00176-f001:**
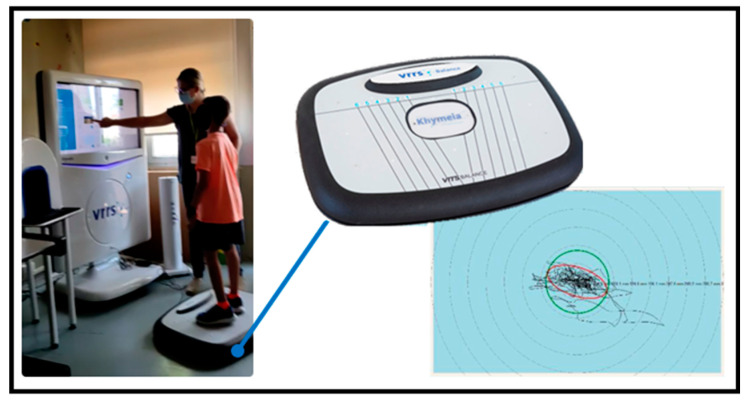
Assessment of postural control by means of VRRS system. On the top left, an example of administering an exercise to a patient according the protocol; in the center, the image of the stabilometric platform used; and on the right the application outputs.

**Figure 2 bioengineering-11-00176-f002:**
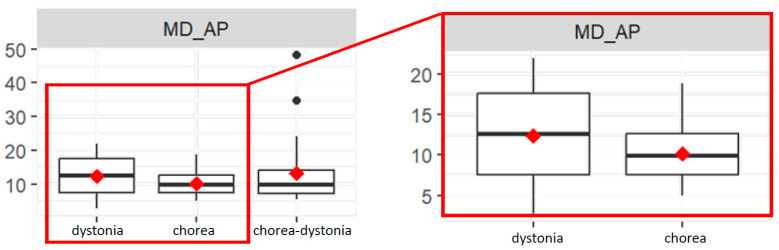
Box plots of mean distance values of the COP in antero-posterior direction (MD-AP) in the three groups (**left**), and focus on the dystonia and chorea groups (**right**).

**Figure 3 bioengineering-11-00176-f003:**
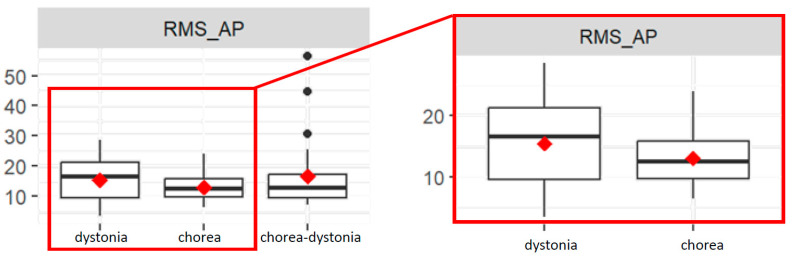
Box plots of root mean square distance values of the COP in antero-posterior direction (RMS_AP) in the three groups (**left**), and focus on the dystonia and chorea groups (**right**).

**Figure 4 bioengineering-11-00176-f004:**
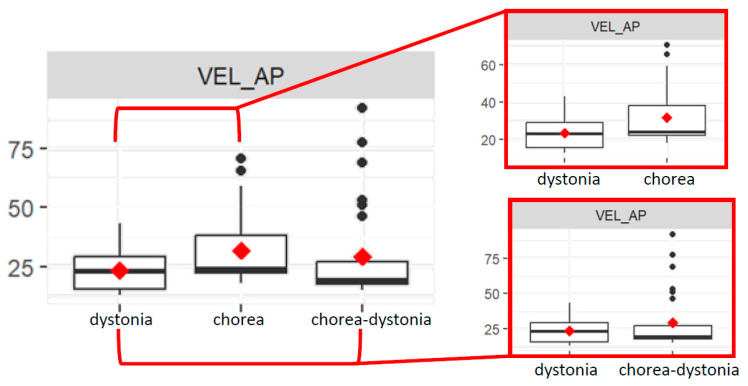
Box plots of average velocity values of the COP in antero-posterior direction (RMS_AP) in the three groups (**left**), and focus on the dystonia and chorea groups (**above right**) and on the dystonia and chorea–dystonia groups (**below right**).

**Figure 5 bioengineering-11-00176-f005:**
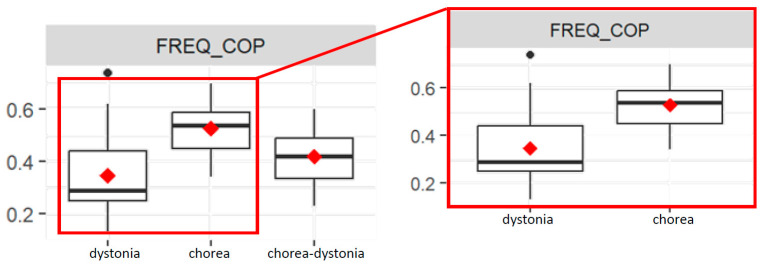
Box plots of average frequency values of the COP (FREQ_COP) in the three groups (**left**), and focus on the dystonia and chorea groups (**right**).

**Figure 6 bioengineering-11-00176-f006:**
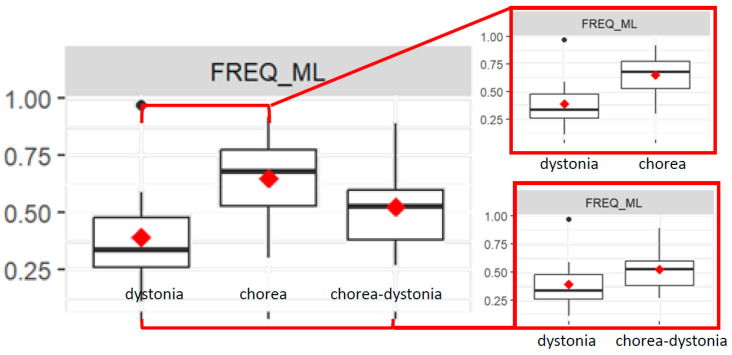
Box plots of average frequency values of the COP in medial–lateral direction (FREQ_ML) in the three groups (**top left**), and focus on the dystonia and chorea groups (**above right**) and on the dystonia and chorea–dystonia groups (**below right**).

**Table 1 bioengineering-11-00176-t001:** COP variables differences among investigated groups.

	Chorea Group vs. Dystonia Group		Chorea–dystonia Group vs. Dystonia Group		Age		Therapy	
	Estimate	*p* Value	Estimate	*p* Value	Estimate	*p* Value	Estimate	*p* Value
MD_COP	−185.64	0.06	−64.01	0.70	−0.95	0.32	113.50	0.13
MD_AP	−128.53	**0.04**	−35.97	0.74	−0.27	0.64	94.56	0.07
MD_ML	−243.20	0.66	10.27	0.47	−0.82	0.31	174.55	0.73
RMS_COP	−246.07	0.73	−9.02	0.65	−1.11	0.30	156.30	0.81
RMS_AP	−156.87	**0.02**	11.93	0.31	−0.37	0.56	114.27	**0.04**
RMS_ML	−322.04	0.63	19.23	0.28	−1.01	0.30	232.18	0.70
ESC_COP	−10,837.86	0.37	5549.68	0.06	−66.60	0.62	9130.21	0.37
ESC_AP	−5649.55	0.49	2479.81	0.17	−47.00	0.61	5185.01	0.46
ESC_ML	−7959.26	0.35	−5361.34	0.75	−33.52	0.72	6571.05	0.36
VEL_COP	−249.00	0.34	−132.86	0.80	−1.15	0.69	159.26	0.46
VEL_AP	−135.91	**0.04**	58.68	**<0.01**	−0.92	0.16	88.34	0.08
VEL_ML	−179.70	0.28	−103.75	0.75	−0.41	0.82	115.30	0.40
SWAY	−3886.73	0.22	−1810.83	0.77	−21.49	0.52	2411.99	0.35
FREQ_COP	2.53	**0.02**	0.32	0.11	0.01	0.39	−1.62	0.05
FREQ_AP	3.62	0.10	0.20	0.61	0.01	0.97	−2.78	0.13
FREQ_ML	2.38	**<0.01**	0.38	**0.02**	0.01	0.30	−1.17	0.05

Caption: MD: mean distance of the COP, AP: anterior–posterior, ML: medium-lateral, RMS: root mean square, ESC: excursion, VEL: velocity, SWAY: sway area, FREQ: frequency.

**Table 2 bioengineering-11-00176-t002:** COP variables interaction among investigated groups, age and therapy.

	Interaction Chorea Group and Age		Interaction Chorea–Dystonia Group and Age		Interaction Chorea Group and Therapy		Interaction Chorea–Dystonia Group and Therapy		Interaction Age and Therapy	
	Estimate	*p* Value	Estimate	*p* Value	Estimate	*p* Value	Estimate	*p* Value	Estimate	*p* Value
MD_COP	18.77	0.07	4.75	0.75	37.67	0.05	77.31	0.06	−16.32	0.1
MD_AP	13.44	0.05	2.72	0.77	2.38	0.06	53.47	0.05	−12.85	0.06
MD_ML	25.32	0.68	−1.71	0.23	37.33	0.48	93.42	0.64	−23.01	0.70
RMS_COP	25.16	0.75	−0.41	0.82	46.34	0.51	102.15	0.69	−22.04	0.78
RMS_AP	16.42	**0.03**	−1.71	0.14	28.81	**0.03**	69.36	**0.02**	−15.53	**0.03**
RMS_ML	33.73	0.64	−2.71	0.12	47.70	0.45	122.87	0.68	−30.60	0.67
ESC_COP	1474.87	0.27	−531.97	**0.05**	737.91	0.75	5511.38	0.29	−1261.76	0.34
ESC_AP	746.91	0.41	−257.13	0.14	943.89	0.57	3302.50	0.36	−717.16	0.43
ESC_ML	1075.80	0.25	472.77	0.76	347.31	0.83	3410.01	0.35	−903.37	0.33
VEL_COP	27.05	0.34	11.40	0.81	47.27	0.37	89.76	0.43	−22.06	0.43
VEL_AP	14.55	**0.04**	−5.88	**<0.01**	22.70	0.07	58.55	**0.04**	−12.29	0.07
VEL_ML	19.71	0.28	9.14	0.76	29.04	0.38	59.95	0.40	−15.84	0.37
SWAY	399.97	0.24	143.63	0.79	811.98	0.20	1517.56	0.26	−341.25	0.31
FREQ_COP	−0.25	**0.03**	−0.02	0.36	−0.52	**0.02**	−1.15	**0.02**	0.23	**0.04**
FREQ_AP	−0.38	0.11	−0.01	0.74	−0.65	0.14	−1.73	0.08	0.38	0.11
FREQ_ML	−0.21	**<0.01**	−0.01	0.38	−0.51	**<0.01**	−1.00	**<0.01**	0.18	**0.02**

Caption: MD: Mean distance of the COP, AP: anterior–posterior, ML: medial–lateral, RMS: Root Mean Square, ESC: excursion, VEL: velocity, SWAY: sway area, FREQ: frequency.

## Data Availability

The data are available upon request to the corresponding author.

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
