# Peer review of "Assessment of Postural Control in Children with Movement Disorders by Means of a New Technological Tool: A Pilot Study"

_bioengineering, 2024, doi:10.3390/bioengineering11020176_

Round 1
Reviewer 1 Report
Comments and Suggestions for Authors
This paper presents an analysis of the outcomes obtained with several postural tools to difference between several movement disorders. This analysis can be useful for developing further machine learning algorithms to obtain biomarkers. But I think, the contribution is very low to be considered for a journal paper. It looks more like a conference paper.
Comments to improve the paper:
- I’d suggest dividing the introduction into introduction and state of the art sections. And include a list of contributions at the end of the introduction.
- You have analyzed many different measures but only few showed significant differences. I’d suggest using a multivariable analysis combining several of these metrics.
- Regarding figures 2-6, I’d suggest not including the zoom strategy. I’d prefer extending the figure to fill the whole page. This way, we can better see the differences.
- Could you provide several classification experiments using machine learning algorithms?
Reviewer 2 Report
Comments and Suggestions for Authors
COMMENTS:
• The work is well written and contains original and interesting research.
• The paper indicates the possibilities of using technology to describe and recognize various movement patterns in children with degenerative diseases, emphasizing the most different movement features.
• A total of 16 children took part in the study - a somewhat small statistical sample.
• The advantages and limitations of the conducted research were indicated.
• This study demonstrated that the use of technology can be a valuable tool in improving understanding of mental illness in children.
• AT THE END, ADD A COMMENT on how the research results help in the treatment of patients.
• In the future, I suggest the authors of the work to establish contact with mathematicians to bring their considerations closer to an in-depth analysis - see. e.g. Lasota, A., Mackey, M.C., Ważewska-Czyżewska, M.: Minimazing theraupetically induced anemia. J. Math. Biol. 13, 149–158 (1981) or Lasota A., Mackey M.C. (1994), Chaos, Fractals, and Noise. Stochastic Aspects of Dynamics. Second Ed., Springer-Verlag, New York.
Reviewer 3 Report
Comments and Suggestions for Authors
This study uses the Virtual Reality Rehabilitation System (VRRS) to evaluate postural control in children with MD. Three different groups of DM were identified according to the prevalent DM: dystonia, chorea, and chorea-dystonia. The study is interesting, since it tries to obtain reliable, objective data on physiological variables of postural control and movement, using a VR platform.
The statistics carried out and the presentation are adequate, as well as the images, which help to understand the differences.
My only suggestions are, on the one hand, to change the title, and put it as a pilot study, since the sample is very small.
Finally, we put more humble conclusions, since the sample is very small, and the variables measured are very few: "This study has demonstrated that the use of technology can be a valuable tool for improving the understanding of MDs in children." Instead of using the verb has shown, this study suggests or presents, or indicates, and so on throughout the conclusion and summary. A little humility for that sample.
Round 2
Reviewer 1 Report
Comments and Suggestions for Authors
The authors have properly addressed my comments. I still think that the contribution is very limited but I have to say that the authors have done an important effort improving the paper.